# Oat (*Avena sativa* L.) Sprouts Restore Skin Barrier Function by Modulating the Expression of the Epidermal Differentiation Complex in Models of Skin Irritation

**DOI:** 10.3390/ijms242417274

**Published:** 2023-12-08

**Authors:** Hyo-Sung Kim, Hyun-Jeong Hwang, Woo-Duck Seo, Sun-Hee Do

**Affiliations:** 1Department of Veterinary Clinical Pathology, College of Veterinary Medicine, Konkuk University, Gwangjin-gu, Seoul 05029, Republic of Korea; 2Division of Crop Foundation, National Institute of Crop Science, Rural Development Administration, Wanju 55365, Republic of Korea

**Keywords:** allergic contact dermatitis, anti-inflammation, epidermal differentiation complex, oat sprout extracts, skin barrier

## Abstract

Oats (*Avena sativa* L.) are used as therapeutic plants, particularly in dermatology. Despite numerous studies on their skin moisturization, anti-inflammation, and antioxidation effects, the precise molecular mechanisms of these effects are only partially understood. In this study, the efficacy of oat sprouts in the treatment of allergic contact dermatitis (ACD) was investigated, and their specific phytoconstituents and exact mechanisms of action were identified. In the in vivo ACD model, by stimulating the mitogen-activated protein kinase signaling pathway, oat sprouts increased the expression levels of proteins associated with skin barrier formation, which are produced during the differentiation of keratinocytes. In addition, in a lipopolysaccharide-induced skin irritation model using HaCaT, steroidal saponins (avenacoside B and 26-deglucoavenacoside B) and a flavonoid (isovitexin-2-o-arabinoside) of oat sprouts regulated the genetic expression of the same proteins located on the adjacent locus of human chromosomes known as the epidermal differentiation complex (EDC). Furthermore, oat sprouts showed immunomodulatory functions. These findings suggest the potential for expanding the use of oat sprouts as a treatment option for various diseases characterized by skin barrier disruption.

## 1. Introduction

Oats (*Avena sativa* L.) have been used for therapeutic purposes since the 12th century, including as dietary supplements and natural remedies for various health conditions [1]. The potent antioxidant effect of oats is derived from avenanthramides and tocopherols, which act as naturally occurring free radical scavengers [2]. Oats, characterized by their high beta-glucan content, contribute to the gradual absorption of nutrients and the reduction in blood cholesterol levels [3]. Additionally, oats serve as a representative plant agent for the treatment of skin diseases. The secretion of anti-inflammatory transforming growth factor β is promoted and dry skin is moisturized through the colloidal extracts of oats, signifying their effectiveness in alleviating conditions of irritated and inflamed skin [4]. Moreover, research has demonstrated that oats can enhance skin moisture levels by modulating the expression of junctional proteins [5,6]. These findings have initiated the development of various cosmetics using oats to improve skin conditions [7]. Researchers have identified various phytochemical ingredients, such as unique polyphenols, avenanthramides, saponins, flavonoids, and high beta-glucans, which are associated with pharmaceutical effects on the human body [1]. However, despite these ongoing studies and the commercialization of oat-based products, the exact molecular mechanism and role of the detailed compounds of oats in treating skin diseases remain only partially understood.

Plant sprouts have garnered attention due to their high levels of health-beneficial phytochemicals and their environmentally friendly characteristics [8]. In a previous study, valuable compounds were extracted and utilized from oat sprouts, with various metabolites isolated from different oat cultivars and identified as glycosylated flavonoids and steroidal saponins [9]. Building upon previous research on oats, we hypothesized that oat sprouts might possess anti-inflammatory properties and could alleviate skin lesions, particularly those requiring moisturization. Therefore, the aim of this study was to investigate the therapeutic effects of oat sprouts and their mechanisms of action in in vivo and in vitro skin irritation models.

Contact dermatitis represents a severe skin condition resulting from exposure to external irritants. Repeated contact with sensitized allergens leads to the development of allergic contact dermatitis (ACD), characterized by intense itching and discomfort. Inflammatory responses and the integrity of the skin barrier are recognized as the principal contributing factors in contact dermatitis [10,11]. ACD primarily occurs owing to the activation of allergen-specific cytotoxic T cells, which are primed by the cutaneous dendritic cell–hapten complex during the sensitization phase. These T cells proliferate, migrate from the lymph nodes into the blood, and recirculate between lymphoid organs [12]. Mast cells play a crucial role as initiators of the initial immune response during the sensitization phase [13]. They are known to facilitate rapid vascular changes that promote the migration of dendritic cells during the initial sensitization phase. Furthermore, the secretion of tumor necrosis factor α (TNFα) by mast cells has a similar effect on cutaneous dendritic cells [14].

Skin barrier function is crucial for maintaining skin homeostasis and health. As keratinocytes in the epidermis undergo differentiation and stratification, they synthesize proteins and lipid molecules to create mechanical and chemical barriers shielding the skin from various irritants. Proteins such as filaggrin, involucrin, and loricrin assemble with outer lipids on the skin surface to form a tough mechanical barrier known as the cornified envelope [15,16]. In addition, these barrier proteins affect the assembly of collagen fibers that provide structural integrity to the skin. Moreover, they break down into moisturizing factors, enhancing skin hydration [17]. Genetic defects in the formation of this barrier have been linked to various skin diseases, such as atopic dermatitis, contact dermatitis, and psoriasis [11,18]. Similarly, several studies demonstrated a correlation between genetic mutations in the filaggrin-coding gene (*FLG*) and increased onset of atopic and ACD [10,19]. Defective skin barriers can exacerbate the progression and prognosis of these diseases by increasing the risk of exposure to allergens and pathogenic microorganisms. Moreover, this inflammatory state triggers the secretion of various chemokines, cytokines, and immune mediators, damaging skin integrity and barrier function and aggravating the situation. Given the significance of the skin barrier function in the onset and progression of skin diseases, enhancing the barrier function may be a crucial therapeutic approach for improving overall skin health.

To validate the therapeutic efficacy of oat sprouts against chronic ACD, an ACD model was induced in SKH-1 mice by subjecting them to repeated oxazolone challenges. This model was chosen to mimic human ACD, characterized by mixed type 1 and type 2 immunity [20]. Subsequently, clinical parameters associated with inflammation and skin barrier function were evaluated at clinical levels following treatment with oat sprout extract (OSE). Additionally, the potential of OSE to regulate the immune response and modulating epidermal barrier function was examined. After confirming the therapeutic efficacy, the focus shifted to elucidating the molecular mechanisms underpinning its effects. Furthermore, experiments were conducted to verify the function and mechanism of the active compounds isolated from oat sprouts using an in vitro model of lipopolysaccharide (LPS)-induced skin irritation with immortalized human keratinocytes (HaCaT).

## 2. Results

### 2.1. Therapeutic Efficacy of OSE on the Oxazolone-Induced ACD Model

As previously described, all mice sensitized with oxazolone exhibited similar gross lesions, regardless of OSE treatment (Figure 1A). All three groups exhibited excoriation and hyperkeratinization on days 14 and 21, respectively. Skin condition scores indicated no significant differences between the treatment groups at the two time points (Figure 1B). Stratum corneum hydration (SCH) also indicated no significant differences between the treatment groups (Figure 1C). However, on day 14, the ACD group displayed relatively high transepidermal water loss (TEWL), suggesting decreased barrier integrity and moisturizing function, while the OSE-treated groups consistently showed lower TEWL values from day 14 onwards. Epidermal and dermal thicknesses showed minimal signs of hyperplasia or thickening (Figure 1D). Microscopic evaluation revealed a reduction in dermal thickness following topical OSE treatment in the oxazolone-induced ACD, while the epidermis thickened after both topical and oral OSE treatment.

Plasma histamine concentrations showed slight fluctuations after OSE treatment, with no significant differences observed between the groups, consistent with the gross lesions. However, the ACD-SPO group exhibited lower plasma IgE levels than the ACD group, and mice subjected to topical OSE treatment without oral administration (ACD-S) demonstrated the lowest plasma IgE and histamine concentrations. The average plasma IgE was nearly three times higher in the untreated ACD group than in the ACD-S group. These results indicate that, despite the absence of visible differences in lesion appearance, the application of oat sprout extract not only contributes to the restoration of skin barrier function disrupted by ACD but also plays a role in reducing IgE, a trigger for inflammation.

**Figure 1 ijms-24-17274-f001:**
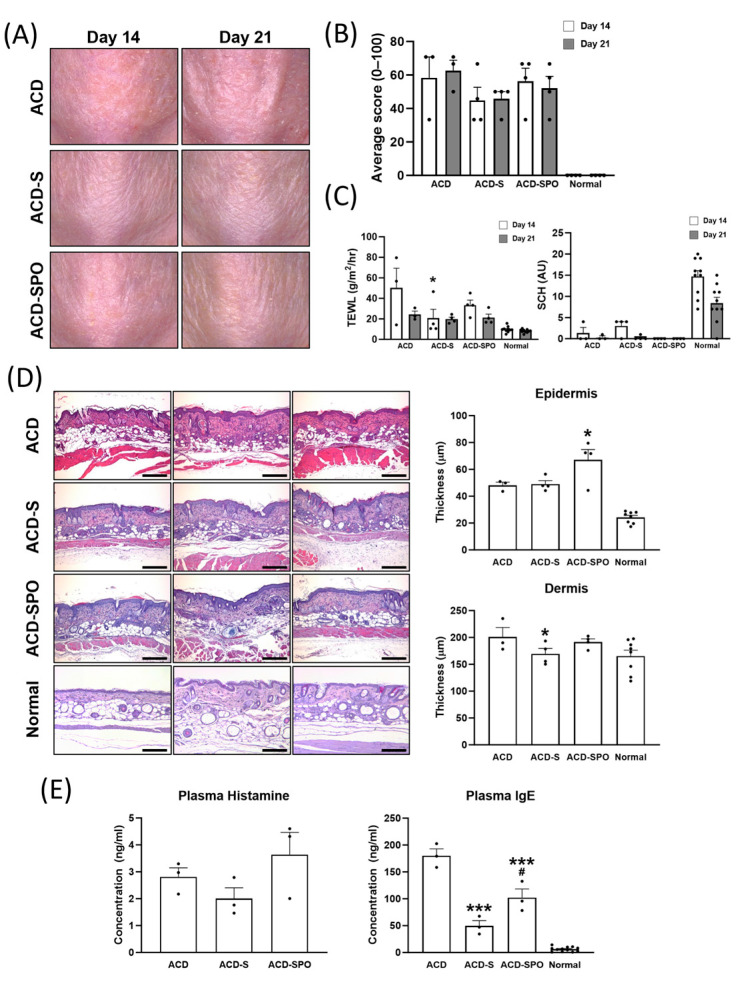
Clinical evaluation of skin lesions in the oxazolone-induced ACD mouse model with OSE treatment. Changes in gross findings of skin lesions in oxazolone-sensitized hairless mice were evaluated with images captured on days 14 and 21 after the first sensitization (**A**). The severity of the lesions was quantified by three blind investigators based on the type and distribution of the lesions (**B**). The scores were assigned on a scale from 0 to 100, with higher scores indicating more severe skin conditions. The effect of OSE treatment on transepidemal water loss (TEWL) and stratum corneum hydration (SCH) were measured on days 14 and 21 (**C**). Histological analysis of the effect of OSE on the oxazolone-induced ACD mouse was performed with the skin tissues from day 21 (**D**). Samples were fixed and stained with hematoxylin and eosin (HE) for the general structure, magnified at 100×. The thickness of the epidermis and dermis was measured with HE-stained slides. Scale bar: 200 μm. Plasma IgE and histamine concentrations were measured using ELISA using blood from the animals on day 21 (**E**). All data are plotted and results are presented as mean with SEM. *, Compared with the ACD group; #, Compared with ACD-S group; *: *p* < 0.05, ***: *p* < 0.001.

### 2.2. OSE Shows Systemic Immune Modulatory Function by Controlling T Lymphocyte Population and Proliferation in the oxazolone-Induced ACD Model

Following the confirmation of reduced serum IgE levels, the evaluation of the systemic immune response, a pivotal component of ACD’s pathological mechanisms, was conducted. Therefore, the splenic mononuclear cell subpopulation was analyzed using flow cytometry. CD4^+^ T lymphocytes, CD8^+^ T lymphocytes, and B lymphocytes were isolated from various cell populations based on the expression of CD45, CD3, CD19, CD4, and CD8a, as illustrated in Figure 2A. A significant reduction in the composition ratio of T lymphocytes, specifically CD4^+^ helper T cells and CD8^+^ cytotoxic T cells, was observed with OSE treatment, accompanied by a notable increase in the population of B lymphocytes (Figure 2C). Importantly, this observed trend remained consistent not only in the ACD-S group but also in the ACD-SPO group. These findings suggest that OSE exhibits systemic immune-modulating effect in skin diseases with an effect on T lymphocyte population, which primarily mediates the immune status of ACD.

### 2.3. OSE Alters Local Cytokine Secretion Levels Depending on the Route of Administration

Subsequently, delving into the local immune system, specifically examining mast cell infiltration and the expression levels of various cytokines in the skin, was performed. Mast cells showed a significantly lower infiltration density after OSE treatment, particularly in the ACD-SPO group (Figure 3A). Among the proinflammatory cytokines (IL-1, IL-5, and IL-6) and the anti-inflammatory cytokine IL-10, only mRNA expression level of IL-5 significantly decreased in the ACD-S group compared to ACD group (Figure 3B). However, there was a tendency toward the decreased expression of proinflammatory cytokines (IL-1 and IL-6) and an increased tendency for IL-10 in the ACD-S group. In contrast, the cytokine expression in the ACD-SPO group showed no significant changes at the genetic or protein level (Figure 3C). The downregulation of IL-5 in the ACD-S group aligned with mast cell infiltration, but this was not observed in the ACD-SPO group, as IL-5 is typically released by mast cells during allergic inflammation. These findings suggest that topical OSE treatment modulates allergic inflammatory reactions in oxazolone-induced ACD by suppressing IgE secretion, subsequently reducing local mast cell recruitment, and consequently, decreasing IL-5 levels in the affected skin area. Additionally, the effect of OSE on immunity is crucially influenced by the route of administration.

### 2.4. Enhancement of Epidermal Barrier Function through the Modulation of Structural Protein Content following OSE Treatment

Recent studies have highlighted that dysfunction in the epidermal barrier system poses the risk of developing severe skin diseases, including ACD and atopic dermatitis. Thus, an analysis of the expression of barrier proteins and their coding genes formed during the stratification and differentiation process of keratinocytes—filaggrin (FLG), loricrin (LOR), and involucrin (IVL)—was conducted. The mRNA expression level of FLG exhibited a decrease in both treatment groups, whereas *IVL* mRNA levels increased in the ACD-SPO group (Figure 4A). In addition, the synthesis of barrier proteins, profilaggrin, loricrin, and involucrin was promoted with topical spray combined with the oral administration of OSE (Figure 4B). Furthermore, these findings were consistent with the results observed in the immunohistochemistry analysis of loricrin (Figure 4C). Along with decreased levels of TEWL, these results suggest that topical and oral OSE treatment can enhance epidermal barrier function and skin moisturization by promoting barrier structural protein synthesis in an oxazolone-induced ACD model.

### 2.5. OSE Modulates Different Cell Signaling Pathways Involved in Skin Proliferation and Differentiation

Stimulation of cell signaling pathways within the skin tissue was evaluated to elucidate the specific mechanism by which OSE affects skin barrier formation. The signal transducer and activator of transcription (STAT) 3 pathway was suppressed upon OSE treatment (Figure 5A,B). In contrast, the expression levels and phosphorylation of signaling proteins associated with the mitogen-activated protein kinase (MAPK) pathway, such as ERK, SAPK, and p38, were investigated using immunoblotting (Figure 5A,C). Their expression and phosphorylation levels increased after OSE treatment and surged particularly with additional oral administration. These results suggest that OSE can biphasically regulate the signaling pathways involved in skin differentiation.

### 2.6. Compounds of Oat Sprouts Mitigated Inflammatory Responses in the HaCaT Irritation Model

To investigate the specific role of the major compounds of oat sprouts in the epidermis, three major compounds of oat sprout were identified: AB, dAB, and IV2A. An in vitro skin irritation model using HaCaT treated with LPS was employed to test their effects.

After pretreatment with LPS, each of the three major compounds was added to the cells. LPS exposure resulted in a 1.5- to 2-fold increase in the relative genetic expression levels of TNFα, IFNG, IL-6, and IL-8 within the cells. However, treatment with the extracted compounds after LPS exposure led to the recovery of IL-6 and IFNG expression levels. Only AB treatment decreased the level of IL-8, while TNFα expression was unaffected by the oat sprout compounds (Figure 6A). Consistent with the genetic analysis result, secretion levels of IL-6, IL-8, TNFα, and IFNG proteins within the cell supernatant were significantly suppressed with the oat sprout compound treatment. While TNFα was not genetically regulated by the compounds, its protein secretion level significantly decreased following treatment with the three compounds when compared with the control (Figure 6B). These results confirm the anti-inflammatory effects of oat sprout and its key compounds, as described in an in vitro skin irritation model. The compounds were able to mitigate the LPS-induced upregulation of pro-inflammatory cytokines and reduce their secretion levels.

### 2.7. Compounds of Oat Sprouts Regulate Epidermal Barrier Protein Expression in a HaCaT Irritation Model

As shown in Figure 7, the effects of the three compounds on keratinocyte differentiation and skin barrier protein production were examined. The results are consistent with the findings observed in the SKH-1 hairless mouse model, indicating the potential of these compounds to affect skin health. Noticeable decreases in filaggrin syntheses were observed following LPS treatment. Treatment with AB and dAB resulted in the recovery of filaggrin expression. Additionally, the three compounds without pretreatment with LPS increased the expression levels of filaggrin compared with that of the normal control group. These results demonstrate that the three active compounds in oat sprouts promote skin barrier protein synthesis within the epidermis and recover the damaged production of these proteins after irritation.

## 3. Discussion

In this study, ACD was induced using oxazolone, a hapten, through initial sensitization, followed by repeated challenges. The pathological features of oxazolone-induced lesions exhibit similar characteristics to the disease observed in daily observations [12]. Using this model, a significant modulation of the immune status in oxazolone-induced ACD lesions was observed upon OSE treatment. Specifically, in the in vivo ACD model, the topical application of OSE led to several notable changes. These changes included a decrease in IgE levels, reduced mast cell infiltration and IL-5 secretion, as well as a decrease in the T cell population within splenocytes. Furthermore, when OSE was administered in combination with both topical and oral routes, an upregulation of the MAPK signaling pathway and increased protein expression levels of filaggrin, loricrin, and involucrin were observed. Additionally, when the three main constituents of OSE, namely AB, dAB, and IV2A, were applied to keratinocytes, reduced expression levels of IL-6, TNF-α, and IFNG were observed. Simultaneously, the protein expression of filaggrin was found to be increased.

Both human and mouse ACD often exhibit a combined type 1 and type 2 immune response [20]. Notably, our results unveiled a decrease in the composition of CD4^+^ and CD8^+^ T-cells following OSE treatment in vivo. Furthermore, the expression of T cell-related cytokines within the skin tissue was modulated. Cytokines associated with the type 1 response, namely IL-8, TNF α, and IFNG, exhibited reductions in the in vitro irritation model after treatment with OSE components. Conversely, cytokines linked to type 2 immunity, such as IL-6 in the in vitro model and IL-5 in the in vivo model, showed decreases. These findings indicate that OSE treatment is effective for ACD, which represents a mixed immune response involving both type 1 and type 2 immunity. Consequently, despite the absence of visible differences, there was a significant reduction in plasma IgE concentration and local mast cell infiltration density. It is plausible that OSE inhibits the priming of systemic hapten-specific T cells in regional lymph nodes by reducing mast cell recruitment during the sensitization phase. However, it is important to note that this study exclusively analyzed the immune status of the mice on day 21 after repeated exposure to oxazolone. Future investigations should delve into the sensitization and elicitation phases over time to establish clear causal relationships.

Skin barrier loss is another significant pathological factor in developing contact dermatitis [11], as it leads to a high chance of exposure to allergens and pathogens, permitting them to penetrate deeper layers of the skin. Various proteins involved in skin barrier formation are encoded by the epidermal differentiation complex (EDC) genes at the 1q21 locus of the human chromosome [21]. Among these, involucrin initially forms bonds with ceramides, and loricrin acts as a crosslinking component, reinforcing this barrier [22]. Filaggrin is an essential component of the epidermal barrier, as it forms a cytofilament and keratin matrix. Moreover, during the final process of filaggrin cleavage, it is cleaved into natural moisturizing factors that comprise up to 10% of the stratum corneum and are reported to be significant skin humectants [21,23]. Therefore, appropriate keratinocyte proliferation and differentiation with the balanced formation and assembly of these proteins constitute the principal part of a healthy skin barrier. 

In this study, OSE significantly increased the expression level of EDC proteins in the oxazolone-induced ACD model, particularly when administered orally. In addition, the expression levels of the structural proteins, loricrin and involucrin as well as the functional filaggrin, were also increased. Similar to previous reports regarding the clinical effect of oats on the skin [6], TEWL in early lesions decreased in our study. This shows that OSE could exhibit skin-protective effects by strengthening skin barrier function in ACD through modulating EDC protein expression.

These proteins are formed during keratinocyte differentiation in the skin. MAPK signaling pathways are involved in the proliferation, differentiation, and inflammatory responses of mammalian cells and comprise three large families, ERK, JNK/SAPK, and p38 [24]. MAPK signaling pathways have been shown to play a significant role in keratinocyte differentiation and EDC expression. Filaggrin decreases with MAPK inhibitors [25], and ERK promotes epidermal proliferation [26]. In this study, the expression of phosphorylated ERK1/2, SAPK/JNK, and p38 consistently increased following OSE treatment, particularly after additional oral administration. These results indicate that the increased expression of EDC proteins by OSE is strongly related to the activation of the MAPK pathway, followed by the promotion of keratinocyte differentiation. Similarly, the STAT3 pathway is involved in skin proliferation, and it reportedly downregulates filaggrin expression [26]. In our model, STAT3 activation decreased after oral administration of OSE, with upregulated filaggrin expression. Therefore, these results suggest that OSE can stimulate the MAPK signaling pathway while simultaneously inhibiting the STAT3 pathway within the skin, modulating EDC protein synthesis, and reinforcing skin barrier formation in ACD. Notably, these events primarily occurred when OSE was orally administered.

Finally, the study aimed to elucidate the roles of various phytochemicals in oat sprouts in mediating their anti-inflammatory and barrier-reinforcing activities. In the in vitro model, the three compounds extracted from oat sprouts showed significant efficacy in inhibiting pro-inflammatory cytokines such as IL-6, IL-8, TNFα, and IFNg, which surged along with LPS-induced damage. Among the three compounds, dAB showed the most pronounced anti-inflammatory effect, decreasing all cytokines oversecreted with LPS. Previous studies have demonstrated that avenanthramides and polyphenols from oats mediate antioxidative and anti-inflammatory functions [27]. Based on these results, it can be inferred that saponins (AB, dAB) and flavonoids (IV2A) are responsible for the anti-inflammatory effects that oat sprouts have on the skin. In addition, considering that some inflammatory cytokines such as TNFα can downregulate EDC [28], the suppression of inflammation by the saponins and flavonoids may be associated with the increased expression of EDC. Together with the in vivo results, these results show that oat sprouts are highly effective in recovering healthy skin after being damaged by irritants. 

These three compounds showed remarkable effects regarding reconstructing damaged skin barrier function in the HaCaT model. The expression levels of filaggrin were significantly diminished after LPS treatment, indicating that LPS greatly disrupted skin barrier function and homeostasis. However, many of these damages were repaired after treatment with the compounds. For example, filaggrin expression significantly recovered with AB and dAB to the level of the non-irritating controls. With filaggrin being the most powerful humectant among barrier proteins [23], our study proposes that the moisturizing effect of oat sprouts originates from the increased filaggrin expressed by its unique steroidal saponin, avenacoside B. Moreover, oat sprouts can be used as a therapeutic reagent for ACD and other diseases such as atopic dermatitis or asthma, which are characterized by a disrupted skin barrier. Verifying the additional function of these chemicals in the purified state to elicit more potent novel therapeutic efficacy in sprout oats is essential.

## 4. Materials and Methods

### 4.1. Preparation of Oat Sprout Extracts 

The plants utilized in this study were prepared following the method described in our previous journal publication [9]. Korean oat cultivars were planted in a growth chamber at the National Institute of Crop Science (NICS), Rural Development Administration (RDA), Jeonbuk, Republic of Korea. The growth conditions were as follows: temperature, 18–20 °C; humidity, 60–70%; illumination intensity, 3300–5500 lux; alternating 9 h of light and 15 h of dark. Harvested seedlings were dried for 2 days at 25 °C and freeze-dried at −78 °C. They were then dissolved at 1 mg/mL in 80% methanol and filtered through a 0.2 um polytetrafluoroethylene syringe filter.

### 4.2. Mouse Model of ACD

Six-week-old female SKH-1 hairless mice were purchased from Orient Bio (Gyeonggi-do, Republic of Korea). Prior to the experiment, all mice were acclimated for 1 week at room temperature under a light–dark cycle. The experimental protocols adhered to in this study received approval from Konkuk University IACUC (KU20005). Skin lesions were induced on the back skins of the mice by topically exposing them to 1% oxazolone (Merck & Co., Rahway, NJ, USA) diluted in acetone on day 0. Starting from 7 days after the initial exposure, the same region of their backs was exposed every other day to 200 μL of 0.1% oxazolone diluted in ethanol until day 21. The first group (ACD) received no treatment except oxazolone sensitization, while the remaining groups (ACD-S and ACD-SPO) were sprayed daily with a 3% total OSE solution from day 7 to day 21. The last group (ACD-SPO) received a daily oral administration of 100 mg/kg OSE for the same duration.

### 4.3. Clinical Evaluation and Sample Collection

TEWL and SCH for each mouse were assessed using the GP Skin Barrier Pro instrument (GP Bio, Gyeonggi-do, Republic of Korea) on day 14 before sensitization and treatment and again on day 21 before euthanasia. Macroscopic changes in the skin were documented using a stereoscopic microscope for clinical evaluation. Clinical scores were determined by three blinded investigators, who considered the severity and distribution of the lesions. The scoring criteria can be found in Appendix A [29]. The animals were euthanized on day 21, and both blood and dorsal skin tissues from the exposed areas were collected for the analysis of cytokine levels, mRNA, and protein levels. Additionally, spleen specimens were collected for the analysis of lymphoid subpopulations.

### 4.4. Histopathological Assessment

Half of the collected skin tissues were fixed in 10% neutral-buffered formalin. Thin slide specimens (4 µm) were prepared through paraffinization and tissue sectioning using a microtome. These sections were then deparaffinized in xylene, rehydrated through an alcohol gradient, and stained with hematoxylin and eosin. Comprehensive microscopic examinations, including measurements of epidermal and dermal thicknesses, were conducted. Mast cells were stained with toluidine blue (pH 2.0). For immunohistochemistry, the hydrated tissue sections were boiled using a TintoRetriever pressure cooker (Bio SB, Goleta, CA, USA) in sodium citrate buffer (pH 6.0) to detect loricrin. For the primary antibody, anti-loricrin (1:200, ab85679, Abcam, Cambridge, UK) was used. The antibody-labeled sections were incubated with secondary antibodies conjugated with Alexa fluor 488 (Abcam). Then, the sections were mounted in 4′-6-diamidino-2-phenylindole (DAPI) containing media (H-1800, Vector Laboratories, Newark, CA, USA) and analyzed under a fluorescence microscope (Leica Microsystems, Wetzlar, Germany). The other half of the skin samples were promptly frozen after sacrifice using liquid nitrogen and stored in a −80 °C deep freezer.

### 4.5. Flow-Cytometry

Splenocytes were obtained from splenic tissues collected during in vivo experiments. The tissues were minced in Dulbecco’s Modified Eagle’s Medium (DMEM, Gibco™, Waltham, MA, USA) supplemented with 10% fetal bovine serum (FBS) (WELGENE Inc., Gyoungsangbuk do, Republic of Korea), and 1% penicillin–streptomycin (Gibco™, MA, USA), and 1X mycoeraser (RDTECH, Busan, Republic of Korea). Subsequently, they were filtered through a 70 µm cell strainer to isolate single lymphoid cells. Furthermore, the samples were gently layered onto Histopaque-1083 (Sigma Aldrich, St. Louis, MO, USA) in a 1:2 ratio and centrifuged for 45 min at room temperature at 400× *g*. The opaque interface was collected and washed twice with Dulbecco’s Phosphate Buffered Saline. Then, 1 mL of Live/Dead working solution from the Fixable Aqua Dead Cell Stain kit (Thermo Fisher, Waltham, MA, USA) was added, and the samples were incubated for 30 min at 4 °C. Next, they were washed, and TruStain FcX Plus (BioLegend, San Diego, CA, USA) was added. After incubating for 10 min at 4 °C, a mixture of surface-staining antibodies was added directly. Samples were incubated for 60 min at 4 °C. Finally, the samples were washed twice with buffer before the flow cytometric analysis. The gating strategy is illustrated in Figure 2. Details about the antibodies used for staining are provided in Appendix A.

### 4.6. Cell Culture and Treatments

HaCaT cells were cultivated in DMEM, supplemented with 10% FBS, 2% penicillin–streptomycin (Gibco™, MA, USA), and 1% Mycoeraser™ (RDTECH, Busan, Republic of Korea) in a humidified chamber at 37 °C with 5% CO_2_. Cells were seeded in 24-well plates at a density of 2.5 × 10^4^ cells/well for real-time PCR and in 6-well plates at a density of 1.25 × 10^5^ cells/well for immunoblotting. Subsequently, the cells were subjected to a 4 h period of serum starvation by incubating with media containing 1% FBS, followed by treatment with 10 µg/mL of LPS (Sigma Aldrich, MO, USA) for 24 h to induce an irritated state. Next, the medium was replaced with DMEM containing 1 µM of dissolved oat sprout compounds, and the cells were incubated for an additional 24 h. The cell supernatants were collected, and further RNA and protein extraction processes were performed.

### 4.7. Real-Time Quantitative Reverse Transcription–Polymerase Chain Reaction (qPCR)

RNA was extracted from the cells and homogenized skin tissues using the RNeasy Mini Kit (QIAGEN, Germantown, MD, USA). cDNA was synthesized using a reverse transcription kit from the same company, excluding genomic DNA. Real-time PCR was conducted using the SYBR Green probe to investigate the effects and mechanisms of oat sprouts on the skin lesions. Each PCR procedure followed these conditions: 95 °C for 10 min, followed by 40 cycles of 15 s at 95 °C, and 1 min at 60 °C. Detailed information about the primers employed in this research is provided in Appendix A.

### 4.8. Immunoblotting

Proteins were obtained from dorsal skin tissue using the Tissue Protein Extraction Reagent (Thermofisher, Waltham, MA, USA). A protease and phosphatase inhibitor cocktail from the same company was added according to the manufacturer’s instructions. Proteins from scraped cells were obtained using the Mammalian Protein Extraction Reagent and inhibitor cocktail from the same company. The concentrations of the extracted protein samples were quantified using the Bradford assay (Bio-Rad, Hercules, CA, USA). Equal amounts of protein were prepared in Laemmli Sample Buffer (Bio-Rad), separated by SDS–polyacrylamide gel electrophoresis, and transferred to nitrocellulose membranes (Merck & Co., CA, USA). They were blocked with a commercial blocking buffer (Bio-Rad) for 5 m at RT, followed by incubation with primary antibodies at 4 °C overnight. Subsequently, the membranes were incubated with secondary antibodies for 2 h at RT. Immunoreactivity was evaluated through densitometric analysis of protein band images using the ImageJ software version 1.51j8 (NIH, Bethesda, MD, USA). Information about the antibodies used for immunoblotting is provided in the Appendix A.

### 4.9. Enzyme-Linked Immunosorbent Assay (ELISA)

Plasma samples from the in vivo experiments and cell supernatants were used to compare the cytokine secretion levels in each group. Commercial ELISA kits for plasma IgE (K3231082, Komabiotech, Seoul, Republic of Korea) and histamine (ab213975, Abcam, Cambridge, UK) concentrations were used for the analysis. In addition, ELISA kits for TNFα (K0331131), IL-8 (K0331216), MCP1(K0331218), and IL-6 (K0331194) were obtained from Komabiotech (Seoul, Republic of Korea) to compare cytokine secretion levels between in vitro groups. The assays were conducted according to the manufacturer’s instructions.

### 4.10. Statistical Analysis

Statistical analyses and graphing were conducted using GraphPad Software Prism V9 (Graphpad Software Inc., La Jolla, CA, USA). Significant differences between groups were assessed using a one-way analysis of variance, followed by Bonferroni’s multiple comparison post hoc test for parametric measures. A significance threshold of *p* < 0.05 was applied.

## 5. Conclusions

This study confirmed the potent activity of oat sprouts as therapeutic agents for ACD and investigated their mechanisms of action. Oat sprouts showed an inhibitory effect on ACD, an anti-inflammatory effect, and a skin barrier reconstruction effect in the ACD models due to modulated cell signaling pathways followed by the recovered expression of EDC proteins. Moreover, specific compounds extracted from oat sprouts were presented, demonstrating notable therapeutic effects on irritated skin. With advanced research on the purification of these chemicals and their application in expanded skin conditions, oat sprouts may effectively cure various intractable diseases in individuals.

## Figures and Tables

**Figure 2 ijms-24-17274-f002:**
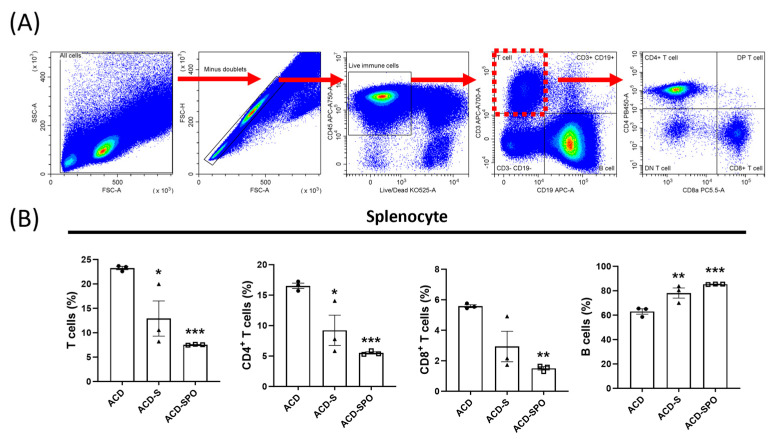
Systemic immune response in the oxazolone-induced ACD mouse model with OSE treatment. A representative gating strategy for flow cytometric analysis using splenocytes is presented (**A**). Lymphocytes were initially separated based on forward and side scatter (FSC and SSC). Following this, CD45^+^ live cells were selected. Among CD45^+^CD3^+^CD19^−^ T cells, CD45^+^CD3^+^CD4^+^CD8^−^ cells were identified as CD4^+^ T cells, and CD45^+^CD3^+^CD4^−^CD8^+^ cells were labeled as CD8^+^ T cells. CD45^+^CD3^−^CD19^+^ cells were identified as B cells. Cell composition ratios among total live splenocytes are presented (**B**). All data are plotted and results are presented as mean with SEM. *, Compared with the ACD group; *: *p* < 0.05, **: *p* < 0.01, ***: *p* < 0.001.

**Figure 3 ijms-24-17274-f003:**
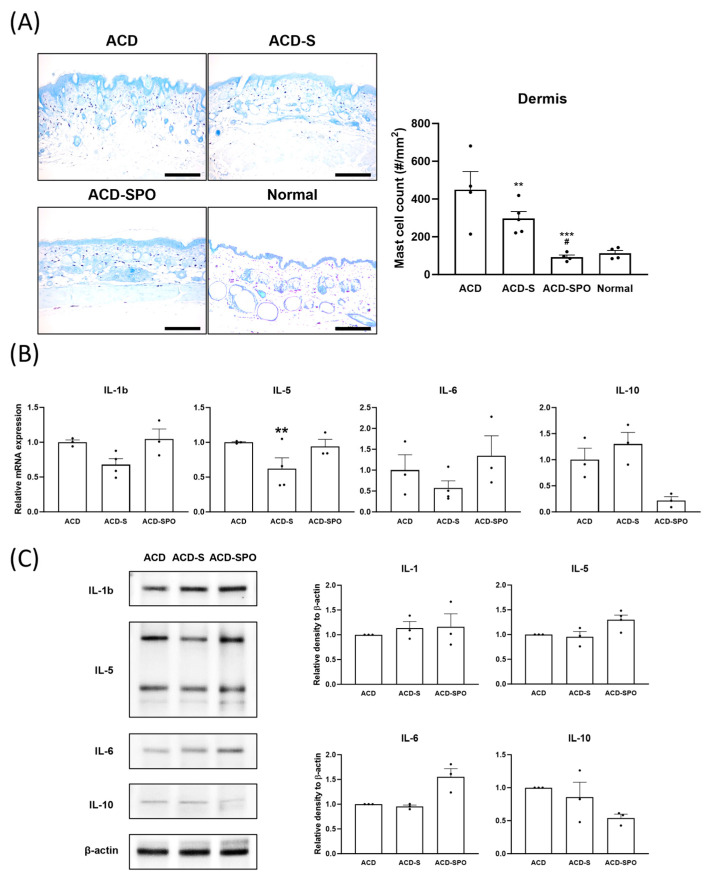
Local immune response in the oxazolone-induced ACD mouse model with OSE treatment. Infiltration density of mast cells was assessed with skin samples stained with toluidine blue, magnified at 100× (**A**). Scale bar: 200 μm. Immune mediator expressions of IL-1, IL-5, IL-6, and IL-10 at the genetic level were analyzed in skin tissues harvested on day 21 using qPCR (**B**). Protein expressions of the same cytokines were analyzed with the tissues using immunoblotting, and quantified densitometry results are presented (**C**). All data are plotted and results are expressed as mean with SEM. *, Compared with the ACD group; #, Compared with ACD-S group; **: *p* < 0.01, ***: *p* < 0.001; #: *p* < 0.05.

**Figure 4 ijms-24-17274-f004:**
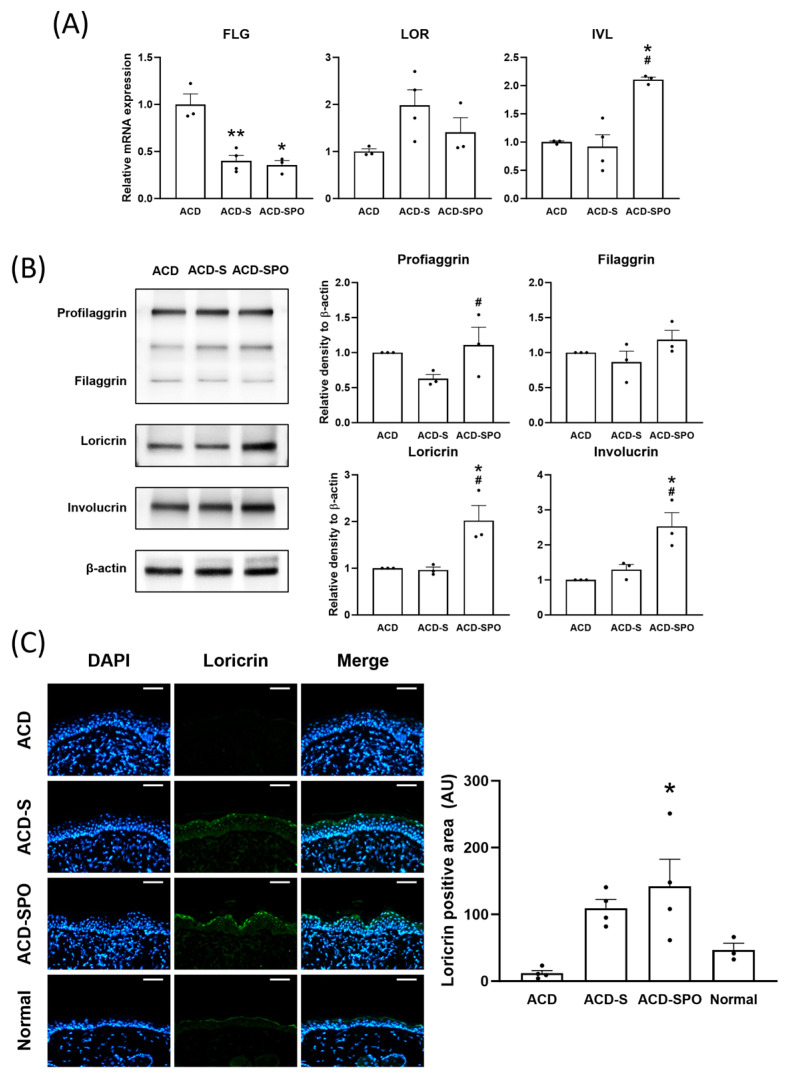
Skin barrier function in the in the oxazolone-induced ACD mouse model with OSE treatment. The expression of genes from EDC was assessed using mRNA isolated from harvested skin tissues treated with or without OSE on day 21 (**A**). Changes in the expression of the same barrier proteins were measured using immunoblotting, and the results were quantified using densitometry (**B**). Differences in the expression of the loricrin were measured using immunohistochemistry, and the results were quantified by a positive area (**C**). Scale bar: 50 μm. All data are plotted and results are presented as mean with SEM. *, Compared with the ACD group; #, Compared with ACD-S group; *: *p* < 0.05, **: *p* < 0.01; # *p* < 0.05.

**Figure 5 ijms-24-17274-f005:**
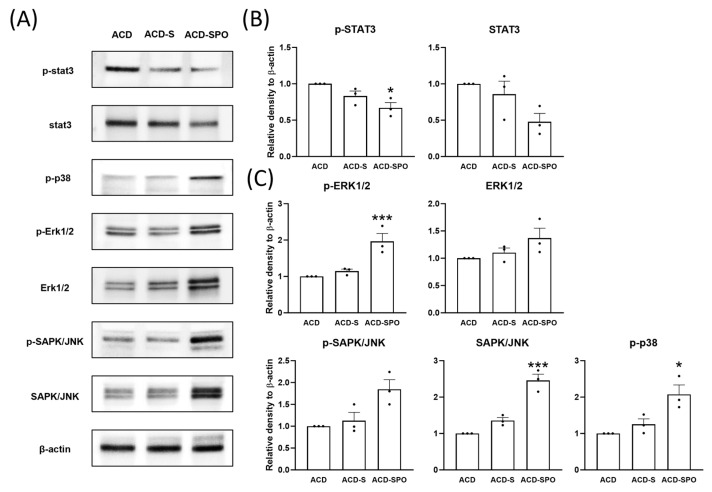
The modulation of cell signaling pathways in the oxazolone-induced ACD mouse model with OSE treatment. The expression levels of signaling proteins in skin samples from day 21 were analyzed using immunoblotting (**A**). Quantitative densitometric analysis was performed for the STAT3 signaling proteins (**B**) and the MAPK signaling proteins (**C**). All data are plotted and results are presented as mean with SEM. Compared with the ACD group *: *p* < 0.05, ***: *p* < 0.001.

**Figure 6 ijms-24-17274-f006:**
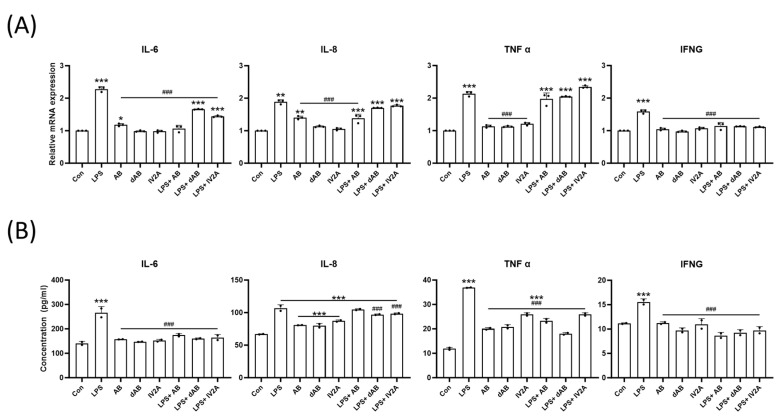
Effect of the compounds of oat sprout on the expression of cytokine in LPS-treated HaCaT. Genetic expression of inflammatory cytokines was quantified with qPCR (**A**). Additionally, the secretion levels of cytokines were analyzed with ELISA within the cell media (**B**). All data are plotted and results are presented as the mean with SD. *, Compared with the control group; #, Compared with the LPS-only group; * *p* < 0.05; **: *p* < 0.01, ***: *p* < 0.001; ### *p* < 0.001.

**Figure 7 ijms-24-17274-f007:**
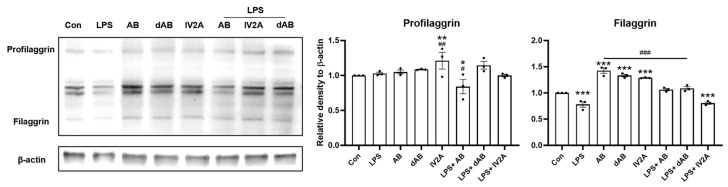
Effect of compounds of oat sprouts on the expression of epidermal barrier proteins in LPS-treated HaCaT. The expressions of profilaggrin and filaggrin were evaluated using immunoblotting, and densitometric results are presented. All data are plotted and results are presented with the mean with SD. *, Compared with the control group; #, Compared with the LPS-only group; *: *p* < 0.05, **: *p* < 0.01, ***: *p* < 0.005; # *p* < 0.05, ## *p* < 0.01, ### *p* < 0.001.

## Data Availability

The data presented in this study are available on request from the corresponding author.

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
