# Peer review of "Oat (Avena sativa L.) Sprouts Restore Skin Barrier Function by Modulating the Expression of the Epidermal Differentiation Complex in Models of Skin Irritation"

_ijms, 2023, doi:10.3390/ijms242417274_

Round 1
Reviewer 1 Report
Comments and Suggestions for Authors
In this manuscript, the authors present the efficacy and mechanism of using oat sprouts to treat skin conditions such as allergic contact dermatitis. The manuscript is well written and, in my opinion, suitable for the IJMS readership. I have a few minor suggestions that the authors might consider, but none of them would prevent moving forward:
1) I recommend supporting the immunoblotting results with imunohistological staining for fillagrin, loricrin and involucrin. This would add more scientific value to the manuscript.
2) Regarding the effects on skin barrier properties, I suggest the authors to also evaluate the barrier function of the HACAT irritation model with the permeability test and/or electrical conductivity.
3) To show the positive effects of oat sprouts on the proliferation of HACAT, the authors could use additional cell-based methods (e.g. MTT assay, LDH assay, etc.) to evaluate the metabolic activity of keratinocytes.
Reviewer 2 Report
Comments and Suggestions for Authors
The authors analyzed the effect of oat sprouts on anti-inflammation and skin barrier function. This paper is well-written and important to elucidate a novel treatment option using this kind of therapeutic plant. I have some comments and questions that are listed below. I hope they will help you revise the manuscript.
1. In Figures 1 to 5, mice without oxazolone-sensitization or data of Day 0 should also be analyzed as normal control to indicate the effect of OSE treatment clearly.
2. In Figure 1, not only TEWL but also SCH (stratum corneum hydration) should also be analyzed to evaluate skin barrier function.
3. In Figures 6 and 7, not only the effect of compounds such as AB, dAB, and IV2A but also the effect of oat sprout on the cytokine and skin barrier protein expression should be should be investigated.
4. The effect of OSE treatment on the skin barrier protein expression should be evaluated in immunohistochemistry using the animal model in Figure 1.
Round 2
Reviewer 2 Report
Comments and Suggestions for Authors
The authors responded to all reviewer’s questions and comments. I don’t have any more comment.